# Application of machine learning approaches to develop predictive models for diabetes and hypertension among Bangladeshi Adults

Gulam Muhammed Al Kibria[1]*, James Ross O'Hagan[2], Golam Shariar[3], Tarina Khan[3], Mohammed Elfaramawi[4]

1 Department of international Health, Johns Hopkins Bloomberg School of Public Health, Baltimore, Maryland, United States of America, 2 US Census Bureau, Baltimore, Maryland, United States of America, 3 New York Institute of Technology College of Osteopathic Medicine, New York City, New York, United States of America, 4 University of Arkansas for Medical Sciences, Little Rock, Arkansas, United States of America

* gkibria1@outlook.com

## Abstract

With rapid urbanization, lifestyle changes, and an aging population, non-communicable diseases (NCDs), including hypertension and diabetes, pose significant public health challenges in Bangladesh and many other low- and middle-income countries. This study used machine learning (ML) approaches to develop predictive models for hypertension and diabetes among Bangladeshi adults. Bangladesh Demographic and Health Survey 2022, a nationally representative cross-sectional survey, data were analyzed. Hypertension was defined as systolic/diastolic blood pressure 140/90 mmHg (or more) or taking any antihypertensive medication. Diabetes was defined as having fasting plasma glucose ≥7.0 mmol/L or using any glucose-lowering drugs. Potential predictors included age, sex, education, wealth quintile, overweight/ obesity, rural-urban residence, and division of residence. Descriptive analysis was conducted, and six ML models were applied: artificial neural network (ANN), random forest, adaptive boosting (AdaBoost), gradient boosting, XGBoost, and support vector machine (SVM). Models' performance and feature importance were reported. We included 13,847 adults (females: 55%). Sensitivity was high across models (up to 0.96 and 0.90 for diabetes and hypertension, respectively). However, the overall specificity was low, particularly for diabetes. The prevalence of diabetes and hypertension was 16.3% and 20.5%, respectively. For diabetes, AdaBoost had the highest AUC (0.699), and SVM had the highest accuracy (0.836); for hypertension, AdaBoost had the greatest AUC (0.775) and accuracy (0.799). Hypertension was the most common diabetes predictor, while overweight/obesity was the most common predictor for hypertension, followed by age and diabetes. Wealth and sex were moderately influential, with education and geographic factors less so. Low specificity across models indicated challenges in identifying non-cases. This ML-driven analysis identified

**Data availability statement:** Data is available from https://www.openicpsr.org/openicpsr/project/241544/version/V2/view.

**Funding:** The authors did not receive any specific funding for the work.

**Competing interests:** The authors have declared that no competing interests exist.

the bidirectional relationship of hypertension and diabetes along with several other predictors, including overweight/obesity, older age, and richer household wealth quintiles. Our findings underscore the need for integrated screening and lifestyle interventions targeting high-risk groups to mitigate future NCD burden.

## Introduction

Non-communicable diseases (NCDs) such as hypertension and diabetes have emerged as critical public health challenges globally, with a particularly alarming rise in low- and middle-income countries (LMICs), including Bangladesh [1–3]. Recent estimates suggest that NCDs are responsible for over two-thirds of global deaths each year, with cardiovascular diseases and diabetes ranking among the top contributors [4,5]. Rapid urbanization, sedentary lifestyles, poor dietary habits, and an aging population have caused the increasing prevalence of these conditions in LMICs, including Bangladesh [1,6,7]. According to the International Diabetes Federation (IDF), approximately 13.1 million adults in Bangladesh were living with diabetes in 2021, a number projected to grow significantly by 2045 if current trends persist [8]. Similarly, hypertension affects nearly one in four adults in the country, often undiagnosed until complications arise [9,10]. These results underscore the urgent need for proactive strategies to identify at-risk populations and implement effective prevention measures.

Artificial intelligence (AI) and machine learning (ML) have revolutionized the analysis of complex health data, and offer a data-driven approach to identify patterns and predictors of diseases or outcomes that are not identifiable by conventional statistical methods [11,12]. For instance, unlike traditional regression models, ML techniques such as decision trees, random forests, and neural networks can handle high-dimensional data and detect non-linear relationships [11].

ML has been successfully applied in various settings to predict burden for conditions like diabetes and hypertension based on factors like body mass index (BMI), family history, and lifestyle habits [13,14]. The lack of data on conditions like hypertension and diabetes was a problem in Bangladesh and other similar LMICs. However, multiple recent surveys and studies have investigated the burden and factors affecting these two conditions [3,6,15,16]. For instance, studies reported that people with older age and higher socioeconomic status (i.e., higher education and wealth) have a higher burden of hypertension/diabetes than those with younger age and lower socioeconomic status [6,16,17]. Despite the growing body of research on NCDs in Bangladesh, several gaps in the literature persist, including the identification of actionable risk factors through advanced analytics like ML [18,19]. This narrow scope overlooks the diversity of risk factors across rural and urban populations, as well as socioeconomic gradients captured in national surveys like the BDHS. Although global studies have highlighted predictors such as obesity and wealth status using ML techniques, there is a lack of evidence on how these factors differ in a Bangladeshi setting. These gaps highlight the need for a comprehensive, ML-driven analysis that uses nationally representative data to provide a holistic understanding of hypertension and diabetes risk.

To overcome these limitations, this study employed ML techniques on the BDHS 2022 dataset. Our objectives were twofold: (1) to identify predictors of diabetes and hypertension among Bangladeshi adults, and (2) to assess the performance of different ML models in this area. This study aimed to reveal new insights into the factors driving these conditions and inform focused public health strategies. The results have the potential to help policymakers develop interventions that tackle the underlying causes of NCDs, thereby reducing their impact on Bangladesh's healthcare system.

## Methods

### Study design

This study analyzed data from BDHS 2022, a nationwide cross-sectional survey in Bangladesh. This survey covered rural and urban regions of all administrative divisions of Bangladesh. It aimed to obtain estimates of major demographic and health indicators, including hypertension and diabetes. Mitra and Associates, a private research firm in Bangladesh, conducted this survey. Approximately two hundred data collectors underwent recruitment and training to conduct interviews. Data collection took place from June to December of 2022 [9]. We analyzed the data for the present study in April 2025.

### Sample design and coverage

BDHS 2022 used a multistage cluster sampling design. It created a sampling frame based on Bangladesh's 2011 housing and population census. This sample frame contained a list of enumeration areas (EA). Households were chosen from the EA in two stages, and data were collected during household visits [9].

For BDHS 2022, 675 EAs were chosen, comprising 425 from rural and 250 from urban areas [10,20]. A higher number of EAs were selected from rural regions to align with the country's population distribution. Approximately 30 households were randomly selected from each EA in the second stage. One-third of the households were eligible for blood pressure and blood sugar measurements. These ensured the representativeness of the samples to ensure reliable estimates of demographic and health indicators. The age eligibility for BDHS 2022 hypertension and diabetes modules was 18. The response rate for the survey was 98%. Detailed information on both BDHS, including survey designs, methodologies, sample size estimations, questionnaires, and results, can be found online [9].

### Outcome variables

The outcome variables of the present study were hypertension and diabetes.

**Hypertension:** This study considered a person hypertensive if s/he met at least one of these three characteristics: systolic blood pressure (SBP) of 140 mmHg or above, diastolic blood pressure (DBP) of 90 mmHg or above, and the participant was getting any hypertensive medication. LIFE SOURCE UA-767 Plus monitors recorded the blood pressure (BP). Three measurements were done with ten minutes of interval between each measurement. Other data were collected during the interval. The mean of the last two BP measurements was used to mark the final BP [9].

**Diabetes:** A person was diabetic if s/he had a fasting plasma glucose of 7.0 mmol/l (or above) or was taking blood sugar-lowering medications. The HemoCue 201+ blood analyzer was used to record the glucose [9].

### Potential factors

Based on scientific plausibility, literature search, and data structure, this study investigated the following variables as potential features associated with the outcomes: age, sex, education level, household wealth quintile, overweight/obesity, and rural-urban place and division of residence. Since diabetes and hypertension have a bidirectional association (i.e., they can increase the risk of one another) [21–23], we investigated hypertension as a risk factor for diabetes and diabetes as a risk factor for hypertension. Multiple demographic and socioeconomic variables were included to minimize potential confounding of associations among predictors and outcomes. The bidirectional inclusion of diabetes and hypertension also helped control for mutual confounding effects.

Age was grouped as 18–34, 35–44, 45–54, 55–64, and 65 or more years. The education level was classified as no formal education, primary (i.e., 1–5 school years), secondary (i.e., 6–10 school years), and college/above education (i.e., 11 or more school years). Respondents reported their current work status. The household wealth index score was obtained by principal component analyses of households' basic building materials (e.g., floors, roofs, and walls), drinking water sources, sanitation facilities, and availability of electricity and other items. Overweight/obesity was defined as body mass index (BMI) ≥25 kg/m² [24]. The score was then stratified into quintiles: poorest, poorer, middle, richer, and richest. Dhaka, Chittagong, Rajshahi, Khulna, Barisal, Rangpur, Sylhet, and Mymensingh were the eight divisions during BDHS 2022 [9].

## Statistical analysis

All analyses were performed using Python (Spyder IDE). First, descriptive analyses were conducted, and the distribution (%) of diabetes and hypertension across the features listed above was reported. Chi-square tests were used to examine the difference in distribution. The dataset was then divided into training (80%) and test sets (20%). As the dataset was imbalanced (i.e., low prevalence of diabetes/hypertension), the synthetic minority over-sampling technique (SMOTE) and adaptive synthetic sampling (ADASYN) were applied to balance the training dataset. SMOTE generates synthetic samples for the minority class by interpolating between existing minority samples [25], and ADASYN does the same but focuses more on harder-to-learn minority samples near decision boundaries [26]. Using both methods allowed us to assess robustness of model performance under different resampling strategies; results presented primarily reflect ADASYN due to its superior sensitivity in preliminary comparisons. Six ML algorithms were implemented for classification: (1) Artificial Neural Network (ANN); (2) Random Forest; (3) AdaBoost; (4) Stochastic Gradient Boosting (GBM); (5) Extreme Gradient Boosting (XGBoost); and (6) Support Vector Machine (SVM).

These six ML algorithms were used to represent diverse and widely used modeling frameworks in epidemiological prediction. Random forest and gradient boosting methods are effective in capturing nonlinear relationships and interactions common in population health data. AdaBoost and XGBoost were included due to their robustness in handling imbalanced datasets and strong performance in health risk prediction. SVM was used for its ability to model complex decision boundaries in high-dimensional feature spaces, while ANN could assess performance under flexible and nonparametric architectures. Overall, these models allow a comprehensive comparison of predictive accuracy, discrimination, and interpretability relevant to public health applications.

Ten-fold cross-validation was applied during training and validation to assess model reliability. Preprocessing to transform categorical variables was performed using one-hot encoding. Model hyperparameters were fine-tuned through a combination of grid search and cross-validation.

Accuracy, sensitivity, specificity, area under the curve (AUC), and F1-score were reported to assess and compare the performance of the six ML models. Receiver operating characteristic (ROC) curves were generated to evaluate the discriminative abilities of the models.

For algorithms capable of estimating feature importance (e.g., random forest, XGBoost, GBM), the most significant predictors were identified using feature importance scores. For linear models such as SVM (with a linear kernel), model coefficients were extracted to determine each variable's contribution to hypertension and diabetes prediction. Bar plots were produced to visually represent the coefficients and feature importance rankings, thereby improving interpretability.

Data manipulation tasks (e.g., recoding or grouping) were carried out using the pandas and numpy libraries. Visualizations were created with Matplotlib and Seaborn, and the ML algorithms were implemented using scikit-learn, XGBoost, and TensorFlow/Keras libraries.

## Results

A total of 13,835 adults were included in the analysis; about 16.3% and 20.6% had diabetes and hypertension, respectively (Table 1). Diabetes affected 15.6% of males and 16.9% of females (P = 0.052). The burden of hypertension was

**Table 1. Distribution of diabetes and hypertension according to features.**

| Variable | Total | Diabetes | | | Hypertension | | |
|---|---|---|---|---|---|---|---|
| | | Yes, 2255 (16.3) | No, 11580 (83.7) | P-value | Yes, 2845 (20.6) | No, 10990 (79.4) | P-values |
| Age (in year) | | | | | | | |
| 18-34 | 5658 | 492(8.7) | 5166(91.3) | <.001 | 411 (7.3) | 5247 (92.7) | <.001 |
| 35-44 | 2957 | 496(16.8) | 2461(83.2) | | 560 (18.9) | 2397 (81.1) | |
| 45-54 | 2124 | 513(24.2) | 1611(75.8) | | 632 (29.8) | 1492 (70.2) | |
| 55-64 | 1680 | 409(24.3) | 1271(75.7) | | 628 (37.4) | 1052 (62.6) | |
| 65 or more | 1416 | 345(24.4) | 1071(75.6) | | 614 (43.4) | 802 (56.6) | |
| Sex | | | | | | | |
| Female | 7635 | 1288(16.9) | 6347(83.1) | .052 | 1793 (23.5) | 5842 (76.5) | <.001 |
| Male | 6200 | 967(15.6) | 5233(84.4) | | 1052 (17.0) | 5148 (83.0) | |
| Education | | | | | | | |
| No education | 3482 | 600(17.2) | 2882(82.8) | .046 | 997 (28.6) | 2485 (71.4) | <.001 |
| Primary | 3504 | 585(16.7) | 2919(83.3) | | 699 (19.9) | 2805 (80.1) | |
| Secondary | 4603 | 694(15.1) | 3909(84.9) | | 756 (16.4) | 3847 (83.6) | |
| College | 2246 | 376(16.7) | 1870(83.3) | | 393 (17.5) | 1853 (82.5) | |
| Wealth quintile | | | | | | | |
| Poorest | 2520 | 250(9.9) | 2270(90.1) | <.001 | 388 (15.4) | 2132 (84.6) | <.001 |
| Poorer | 2715 | 328(12.1) | 2387(87.9) | | 483 (17.8) | 2232 (82.2) | |
| Middle | 2643 | 399(15.1) | 2244(84.9) | | 507 (19.2) | 2136 (80.8) | |
| Richer | 2880 | 512(17.8) | 2368(82.2) | | 618 (21.5) | 2262 (78.5) | |
| Richest | 3077 | 766(24.9) | 2311(75.1) | | 849 (27.6) | 2228 (72.4) | |
| Place of residence | | | | | | | |
| Rural | 9113 | 1231(13.5) | 7882(86.5) | <.001 | 1747 (19.2) | 7366 (80.8) | <.001 |
| Urban | 4722 | 1024(21.7) | 3698(78.3) | | 1098 (23.3) | 3624 (76.7) | |
| Division of residence | | | | | | | |
| Barishal | 1486 | 214(14.4) | 1272(85.6) | <.001 | 286 (19.2) | 1200 (80.8) | <.001 |
| Chattagram | 1946 | 357(18.3) | 1589(81.7) | | 400 (20.6) | 1546 (79.4) | |
| Dhaka | 1798 | 379(21.1) | 1419(78.9) | | 347 (19.3) | 1451 (80.7) | |
| Khulna | 1773 | 294(16.6) | 1479(83.4) | | 397 (22.4) | 1376 (77.6) | |
| Mymensingh | 1610 | 180(11.2) | 1430(88.8) | | 275 (17.1) | 1335 (82.9) | |
| Rajshahi | 1773 | 263(14.8) | 1510(85.2) | | 420 (23.7) | 1353 (76.3) | |
| Rangpur | 1747 | 284(16.3) | 1463(83.7) | | 392 (22.4) | 1355 (77.6) | |
| Sylhet | 1702 | 284(16.7) | 1418(83.3) | | 328 (19.3) | 1374 (80.7) | |
| Diabetes | | | | | | | |
| Yes | 2258 | | | | 773 (34.3) | 1482 (65.7) | <.001 |
| No | 11589 | | | | 2072 (17.9) | 9508 (82.1) | |
| Hypertension | | | | | | | |
| Yes | 2845 | 773(27.2) | 2072(72.8) | <.001 | | | |
| No | 10990 | 1482(13.5) | 9508(86.5) | | | | |
| Overweight/obesity | | | | | | | |
| Yes | 3798 | 904(23.8) | 2894(76.2) | <.001 | 1234 (32.5) | 2564 (67.5) | <.001 |
| No | 10037 | 1351(13.5) | 8686(86.5) | | 1611 (16.1) | 8426 (83.9) | |

significantly higher in females (23.5%) compared to their male counterparts (17.0%; P<0.001). Prevalence of both conditions increased with age (P<0.001), peaking at 24.4% for diabetes (65+years) and 43.3% for hypertension (65+years). Education showed a modest association with diabetes (P=0.046), while hypertension was highest among those with no education (28.6%; P<0.001). Wealth quintiles revealed a gradient, with diabetes, increased from 9.9% among the poorest to 24.9% among the richest and hypertension from 15.4% (poorest) to 27.6% (richest) (P<0.001). Urban residents had higher rates (diabetes: 21.7%; hypertension: 23.3%) than rural residents (13.5%; 19.2%; P<0.001). Divisionally, Dhaka had the highest diabetes prevalence (21.1%), while Rajshahi had the highest prevalence for hypertension (23.6%; P<0.001). Overweight/obesity significantly increased both conditions' prevalence (P<0.001).

Table 2 shows the performance metrics of six ML models. For diabetes, XGBoost achieved the highest accuracy (0.821) and F1 score (0.899), with a sensitivity of 0.958, indicating strong positive class detection, though specificity was low (0.127). Gradient boosting followed closely (accuracy: 0.803, F1: 0.885). SVM and random forest also performed well, with accuracies of 0.805 and 0.790, respectively. AdaBoost had the lowest accuracy (0.682) but a higher AUC (0.694). For hypertension, XGBoost again outperformed others (accuracy: 0.779, F1: 0.865), with a sensitivity of 0.899. Gradient boosting and SVM showed balanced performance (accuracies: 0.751, 0.755). AdaBoost had the highest AUC (0.770) but lower accuracy (0.700). Overall, XGBoost consistently demonstrated superior predictive performance across both conditions, particularly in sensitivity and F1 score.

In Fig 1, ROC curves to evaluate the discriminative ability of six classification models were presented. For diabetes, gradient boosting achieved the highest AUC (0.695), followed closely by AdaBoost (0.694), indicating strong overall performance in distinguishing classes. XGBoost (AUC: 0.652) and SVM (0.632) showed moderate performance, while ANN had the lowest AUC (0.585). Random forest (AUC: 0.636) performed slightly better than ANN but lagged behind the top models. For hypertension, Gradient Boosting again led with the highest AUC (0.773), closely followed by AdaBoost (0.770) and XGBoost (0.745), reflecting robust classification capabilities. SVM (AUC: 0.737) and random forest (0.704) showed moderate performance, while ANN had the lowest AUC (0.680). Across both conditions, Gradient boosting consistently demonstrated superior discriminative power, with AdaBoost also performing notably well, particularly in AUC metrics.

The feature importance for predicting diabetes was reported in Table 3. We reported the features that were among the top 5 for 6 models. Notably, the top 5 features for both outcome variables were similar. For diabetes, hypertension ranked first across all models (frequency=6), followed by overweight/obesity (second, frequency=6) and age (third, frequency=6). Wealth quintile and sex ranked fourth–fifth, while education, division, and place of residence ranked

**Table 2. Performance metrices of the evaluation models obtained by ADASYN.**

| Outcome | Model | Accuracy | AUC | Precision | Sensitivity | Specificity | F1 Score |
|---|---|---|---|---|---|---|---|
| **Diabetes** | ANN | 0.759 | 0.585 | 0.857 | 0.854 | 0.284 | 0.856 |
| | Random Forest | 0.790 | 0.636 | 0.848 | 0.911 | 0.179 | 0.879 |
| | AdaBoost | 0.682 | 0.694 | **0.891** | 0.706 | **0.563** | 0.787 |
| | Gradient Boosting | 0.803 | **0.695** | 0.863 | 0.909 | 0.271 | 0.885 |
| | XGBoost | **0.821** | 0.652 | 0.847 | **0.958** | 0.127 | **0.899** |
| | SVM | 0.805 | 0.632 | 0.859 | 0.916 | 0.245 | 0.887 |
| **Hypertension** | ANN | 0.683 | 0.680 | 0.850 | 0.726 | 0.526 | 0.783 |
| | Random Forest | 0.750 | 0.704 | 0.830 | 0.858 | 0.349 | 0.844 |
| | AdaBoost | 0.700 | 0.770 | **0.899** | 0.698 | **0.711** | 0.786 |
| | Gradient Boosting | 0.751 | **0.773** | 0.876 | 0.797 | 0.582 | 0.835 |
| | XGBoost | **0.779** | 0.745 | 0.833 | **0.899** | 0.333 | **0.865** |
| | SVM | 0.755 | 0.737 | 0.862 | 0.820 | 0.514 | 0.840 |

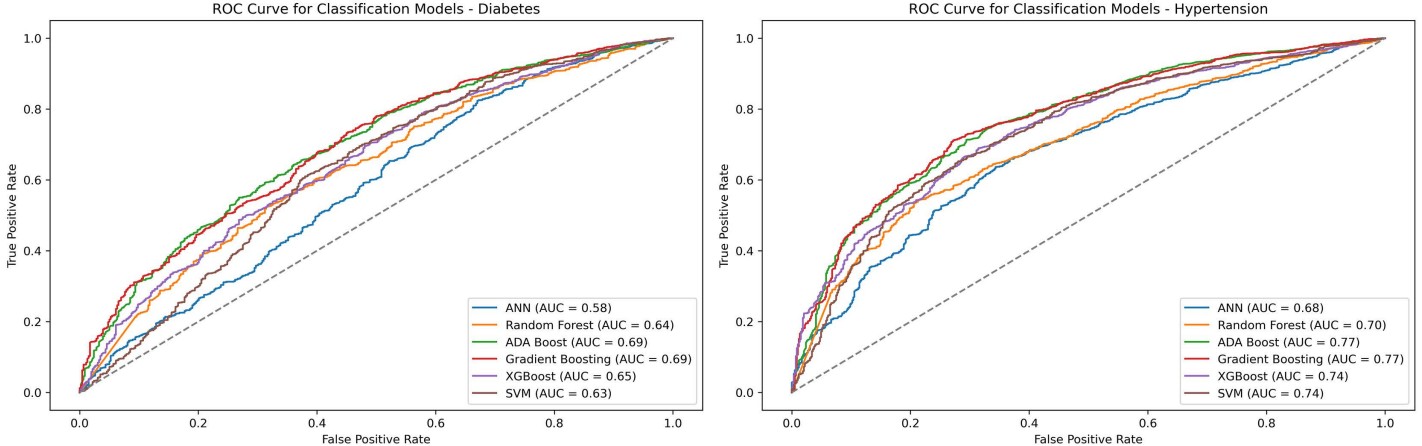

**Fig 1. Receiver operating characteristics curve for the outcomes.**

**Table 3. The top 5 identified features in all studied models.**

| Features | Frequency as top 5 features among 6 models | Rank in Particular Model | | | | | |
|---|---|---|---|---|---|---|---|
| | | ANN | Random Forest | AdaBoost | Stochastic Gradient Boosting | XGBoost | SVM |
| **Diabetes** | | | | | | | |
| Hypertension | 6 | 1 | 1 | 1 | 1 | 1 | 1 |
| Overweight/Obesity | 6 | 2 | 2 | 2 | 2 | 2 | 2 |
| Age | 6 | 3 | 3 | 3 | 3 | 3 | 3 |
| Wealth Quintile | 6 | 4 | 4 | 4 | 4 | 4 | 5 |
| Sex | 6 | 5 | 5 | 5 | 5 | 5 | 4 |
| Education | 0 | 6 | 7 | 6 | 7 | 7 | 6 |
| Division | 0 | 7 | 6 | 7 | 6 | 6 | 7 |
| Place of residence | 0 | 8 | 8 | 8 | 8 | 8 | 8 |
| **Hypertension** | | | | | | | |
| Overweight/obesity | 6 | 1 | 1 | 1 | 1 | 1 | 1 |
| Age | 6 | 2 | 2 | 2 | 2 | 2 | 2 |
| Diabetes | 6 | 3 | 3 | 3 | 3 | 3 | 3 |
| Wealth Quintile | 6 | 4 | 4 | 4 | 4 | 4 | 5 |
| Sex | 6 | 5 | 5 | 5 | 5 | 5 | 4 |
| Education | 0 | 6 | 7 | 6 | 7 | 7 | 6 |
| Division | 0 | 7 | 6 | 7 | 6 | 6 | 7 |
| Place of residence | 0 | 8 | 8 | 8 | 8 | 8 | 8 |

sixth–eighth. For hypertension, overweight/obesity was the top predictor (first, frequency = 6), followed by age (second, frequency = 6) and diabetes (third, frequency = 6). Wealth quintile and sex ranked fourth–fifth, with education, division, and place of residence at sixth–eighth. Health-related features (hypertension, overweight/obesity, and diabetes), and age consistently dominated. Education and geographic factors showed lower importance, with SVM slightly varying in ranking wealth quintile and sex. These findings highlight key risk factors for diabetes and hypertension, informing targeted interventions.

## Discussion

This study used nationally representative data from BDHS 2022 and applied modern ML approaches to identify and rank predictors of diabetes and hypertension among Bangladeshi adults. In addition to observing the bidirectional relationship of diabetes and hypertension, age, wealth quintile, and overweight/obesity were potential associated features for the studied conditions in Bangladesh. Our findings not only confirmed known risk factors for diabetes or hypertension but also provided insights into the demographic, socioeconomic, and health-related variables that impact these two outcomes in this country. These insights have important implications for targeted prevention, policy development, and health system planning.

From a methodological perspective, the use of ML algorithms allowed for robust predictive modeling. The identification of complex relationships among variables might have been overlooked using traditional statistical techniques. AdaBoost, XGBoost, and gradient boosting consistently outperformed other models in terms of sensitivity and F1 scores for both conditions. Their superior performance demonstrated the value of ensemble learning methods in health prediction tasks, specifically with multidimensional data [11,12]. SVM and ANN also performed well, though SVM had the lowest AUC for diabetes, indicating weaker discriminative ability for that outcome. The use of ADASYN likely contributed to the high sensitivity observed across models by emphasizing difficult-to-classify minority cases. However, this may have occurred at the expense of specificity and underscored the trade-off inherent in resampling strategies for low-prevalence outcomes. Moreover, the observed low specificity across models, particularly for diabetes, could result from the low prevalence of both outcomes (i.e., 16% for diabetes and 21% hypertension). These low specificity levels suggest that further model refinement and perhaps inclusion of additional behavioral or clinical variables may enhance performance [11,12].

Compared with traditional regression models commonly used in epidemiology, ML approaches offered advantages in capturing nonlinear relationships and complex interactions without prespecified functional forms. Although logistic regression remains valuable for inference, the models used here demonstrated improved discrimination and ranking of predictors [11,12]. These findings highlight the complementary role of ML in population health analytics.

The feature importance rankings revealed a consistent pattern across all ML models. Health-related variables (i.e., hypertension, diabetes, and overweight/obesity) and age were the top predictors. Wealth quintile and sex also had a significant association; however, education, place of residence, and division had relatively lesser influence [6,16,27]. This hierarchy of predictors reinforces the notion that while demographic and geographic contexts matter, physiological and lifestyle-related indicators are the most immediate determinants of NCD risk [16,28].

The prevalence estimates by features observed in this study underscore the significance of known predictors that could be used to develop and implement prevention and control programs in the context of this country. For instance, consistent with prior research, both conditions showed a strong association with older age, and individuals aged 65 years and above have the highest burden [6,27,29]. This age-associated rise is consistent with biological aging processes and cumulative exposure to risk factors such as obesity, poor diet, and sedentary behavior [18]. Although women had a higher burden of hypertension, the diabetes burden was not significantly higher. These sex-related differences may reflect differences in healthcare access, behavioral risk profiles, and sociocultural dynamics [30,31]. The observed associations align with findings from prior studies in Bangladesh and other LMICs [27,29,30,32,33], which similarly reported increasing diabetes and hypertension prevalence with age, higher socioeconomic status, and urban residence. These studies also highlighted obesity, wealth, and urbanization as consistent predictors of cardiometabolic conditions, supporting the generalizability of these findings across South Asian populations.

Among the most important predictors across all six ML models were hypertension for diabetes (and vice versa). This finding reaffirmed the bidirectional and interdependent relationship between diabetes and hypertension [21,22]. Previous studies have documented that hypertension increases insulin resistance, and diabetes can impair vascular function, which ultimately increases blood pressure levels [21,22,31]. Our findings support integrating screening and management protocols for both conditions simultaneously, especially in primary and outpatient care settings in upazila (i.e., sub-district) health complexes, district hospitals, or tertiary (e.g., medical college) hospitals.

Overweight/obesity appeared as a predictor for both diabetes and hypertension across all models. This echoed global evidence on the central role of excess adiposity in metabolic dysfunction [34,35]. Cardiometabolic syndromes can also occur due to adipocyte hypertrophy, visceral adiposity, and ectopic fat deposition. Additionally, compensatory insulin secretion is closely associated with obesity [36]. The transition from pre-diabetes to type 2 diabetes is caused by this hypersecretion, which also causes insulin resistance and ultimately β-cell failure [35,36]. The overall prevalence of overweight/obesity has increased in Bangladesh. For instance, the prevalence of overweight/obesity was 25% among 35+-year-old people in 2011; this increased to 32% in 2022 [9,20]. The rising prevalence has been driven by urbanization, dietary transitions, and physical inactivity and poses a significant challenge to the ongoing burden of NCDs like diabetes and hypertension [1,17]. Public health campaigns promoting healthy lifestyles, including physical activity and nutritional education, are urgently needed to curb this trend.

Wealth quintile, a proxy for socioeconomic status, demonstrated a "dose-response relationship" with both conditions; disease prevalence increased from the poorest to the richest households. Another predictor variable, education level, often considered as a socioeconomic variable in addition to wealth quintile, did not have a significant association with either of the conditions. This is contrary to previous research and suggests that the significance of education level on the burden of hypertension/diabetes is declining [16,27]. The wealth pattern likely reflects a shift in the burden of diabetes/hypertension toward wealthier segments of the population due to their greater access to energy-dense diets, sedentary occupations, and urban environments. However, this does not diminish the vulnerability of populations with lower household wealth, who may face more barriers to receiving early diagnosis and effective treatment [37]. Socioeconomic gradients in NCD prevalence call for a dual approach to prevent or control these conditions. It signifies targeting high-risk behaviors in affluent groups while ensuring equitable access to care for the poor.

Our study fills critical gaps in the Bangladeshi literature on NCDs. Unlike earlier studies that primarily focused on prevalence estimates, predictive analytics to identify actionable risk factors were employed [6,19,27,29]. Additionally, by incorporating ML techniques into nationally representative data analysis, this study moved beyond traditional epidemiological approaches, and contributed a more nuanced and dynamic understanding of disease risk. This methodology also demonstrated the potential for future integration into health decision-support systems, enabling personalized risk assessments and real-time public health surveillance.

High sensitivity with low specificity suggests that the models are effective for screening and identifying high-risk individuals but may overclassify non-cases. From a public health perspective, this trade-off may be acceptable in early detection programs where missing true cases is costlier than false positives. Future work could improve specificity by incorporating behavioral variables, continuous biomarkers, cost-sensitive learning, probability threshold optimization, or ensemble calibration techniques. These ML-derived risk profiles can inform targeted screening strategies within Bangladesh's primary healthcare system, particularly for older adults, overweight individuals, and wealthier urban populations. Integration of such predictive tools into routine health surveys or electronic health systems could enable risk stratification and efficient allocation of limited preventive resources.

Limitations of this study also merit discussion. First, the cross-sectional design of BDHS restricts causal inference. Second, some important behavioral and clinical predictors (e.g., dietary intake, physical activity levels, and family history) were not available in the dataset, and that may have impacted the accuracy of the models. Third, the models were trained on structured categorical variables; incorporating continuous or unstructured data in future analyses may yield improved performance. Lastly, although ML offers powerful tools for prediction, the black-box nature of some algorithms (e.g., ANN) may hinder interpretability for clinical applications.

## Conclusion

This study demonstrates that survey data with advanced ML models can effectively identify key predictors of hypertension and diabetes. Our results highlight the dominant role of modifiable risk factors such as overweight/obesity and wealth, in

addition to the reinforcing relationship between the two conditions. These findings may help guide the design of targeted, data-driven public health interventions that prioritize high-risk groups, promote early screening, and support healthy lifestyle changes to reduce the burden of NCDs across Bangladesh.

## Author contributions

**Conceptualization:** Gulam Muhammed Al Kibria.

**Formal analysis:** Gulam Muhammed Al Kibria, James Ross O'Hagan.

**Investigation:** Gulam Muhammed Al Kibria, James Ross O'Hagan, Golam Shariar, Tarina Khan, Mohammed Elfaramawi.

**Methodology:** Gulam Muhammed Al Kibria, Tarina Khan, Mohammed Elfaramawi.

**Software:** Gulam Muhammed Al Kibria.

**Writing – original draft:** Gulam Muhammed Al Kibria, James Ross O'Hagan.

**Writing – review & editing:** Golam Shariar, Tarina Khan, Mohammed Elfaramawi.

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
