## [Decision Letter · Decision Letter 0]

17 Dec 2025

PGPH-D-25-01388

Application of Machine Learning Approaches to Develop Predictive Models for Diabetes and Hypertension among Bangladesh Adults

Dear Dr. Kibria,

Thank you for submitting your manuscript to PLOS Global Public Health. After careful consideration, we feel that it has merit but does not fully meet PLOS Global Public Health’s publication criteria as it currently stands. Therefore, we invite you to submit a revised version of the manuscript that addresses the points raised during the review process.

Abstract and throughout: remove the term 'rates' when mentioning prevalence, prevalence is not a rate.

Potential factors: You included both male and female, please clarify why you mentioned only women's age?

Statistical analysis: "We used chi-square tests to examine the association of these factors with hypertension and diabetes." Chi-square test does not examine the association. It is used to see the differences in the distribution. Please revise the sentence accordingly.

If you think that some reviewers' comments are not relevant to your manuscript, respond to them accordingly.

We look forward to receiving your revised manuscript.

Kind regards,

Saifur R. Chowdhury, BScN, MPH, PhD (c)

Academic Editor

Journal Requirements:

1. Please note that PLOS Global Public Health has specific guidelines on code sharing for submissions in which author-generated code underpins the findings in the manuscript. In these cases, all author-generated code must be made available without restrictions upon publication of the work. Please review our guidelines at https://journals.plos.org/globalpublichealth/s/materials-and-software-sharing#loc-sharing-code and ensure that your code is shared in a way that follows best practice and facilitates reproducibility and reuse.

Additional Editor Comments (if provided):

Reviewers' comments:

Reviewer's Responses to Questions

**Comments to the Author**

1. Does this manuscript meet PLOS Global Public Health’s publication criteria?

Reviewer #1: Partly

Reviewer #2: Yes

Reviewer #3: Yes

Reviewer #4: Yes

2. Has the statistical analysis been performed appropriately and rigorously?

Reviewer #1: No

Reviewer #2: Yes

Reviewer #3: Yes

Reviewer #4: Yes

3. Have the authors made all data underlying the findings in their manuscript fully available (please refer to the Data Availability Statement at the start of the manuscript PDF file)?

Reviewer #1: No

Reviewer #2: Yes

Reviewer #3: Yes

Reviewer #4: Yes

4. Is the manuscript presented in an intelligible fashion and written in standard English?

Reviewer #1: Yes

Reviewer #2: Yes

Reviewer #3: Yes

Reviewer #4: Yes

Reviewer #1: This manuscript contributes valuable evidence on body image satisfaction and related psychosocial factors among university students in a low- and middle-income country setting. The topic aligns well with global public health priorities concerning mental health, self-perception, and youth well-being.

Strengths:

Large, multi-institutional sample with strong ethical compliance.

Use of validated measurement instruments.

Rich discussion linking findings to regional and global literature.

Revisions Needed:

Clarify measurement procedures and justify BAS-2 categorization; report mean, SD, and range.

Add reliability statistics for the instruments used.

Reconcile numerical inconsistencies between results text and tables.

Confirm regression diagnostics and check for multicollinearity.

Add a note on referral or support options for participants with high PHQ-4 distress scores.

Deposit anonymized data in a public repository to meet open data standards.

Condense and refocus the discussion for clarity.

Overall Recommendation:

Major Revision — The paper is strong in concept and ethics, but the statistical and reporting details require correction and expansion before publication.

Reviewer #2: The authors have provided a strong manuscript with clear research questions and robust methodology. I am satisfied with the clarity and presentation of the results. The manuscript has provided expected answers to the research questions raised.

Reviewer #3: The manuscript addresses an important public health issue using contemporary machine learning (ML) methods on nationally representative data. The study is well-motivated, methods are generally sound, and findings are relevant for policy and prevention. However, several areas require clarification, methodological justification, and improved reporting to meet the journal’s standards.

The tenses in the work are not consistent, kindly make all tenses uniform within the manuscript

Comments

The authors applied six ML models but did not clearly justify why these specific algorithms were chosen over others. A brief rationale for each model’s relevance to epidemiological prediction would strengthen the methodological rigor.

While SMOTE and ADASYN were used, the rationale for applying both and their differential impact on model performance is not discussed. The manuscript should briefly explain why these methods were chosen and how they affected model outcomes, especially given the

The models show high sensitivity but very low specificity (e.g., 0.127 for diabetes in XGBoost), indicating poor ability to correctly identify non-cases. The authors should discuss the public health implications of this and suggest ways to improve specificity in future work.

The study would benefit from a comparison with traditional statistical models to contextualize the added value of ML in this setting.

The abstract mentions “low specificity across models” but does not provide values. Consider including key specificity figures for transparency.

BMI cutoffs for overweight/obesity should be specified (e.g., ≥25 kg/m²) for international readability.

The discussion could better highlight how these ML findings translate to actionable public health interventions in Bangladesh

Check consistency in terminology (e.g., “AdaBoost” vs. “ADA Boost” in Table 2).

The tenses in the work are not consistent, kindly make all tenses uniform within the manuscript

Kindly go through the entire work and ensure all grammatical errors are checked. Example w “prevalce of both outcomes” in the discussion should be checked

Reviewer #4: I commend the Authors for their choice of these important NCDs of public health importance, sample size is large enough, methodology is novel and conclusion is well supported by data , however may I call the attention of the Authors to these in order to improve the manuscript.

1) Tenses of reportage in manuscript is expected to be past in form, this is however not the case in the entire manuscript for instance the statement “we aim to……” as seen in the last paragraph of the introduction section was not in past form.

2)Please avoid personalization of action in the manuscript, e.g “We …….” As seen severally in the manuscript particularly in the data analysis and discussion sections , it could be better to use passive word such as analysis was done.

3)It may not be out of place for the Authors to compare the research findings with the outcomes of similar studies from other places as no research is an island of its own.

4) In the recommendation please use persuasive and not an obligatory tone for instance it’s not ideal to use “Should” while making recommendations as seen in the conclusion section of the manuscript.

5)Was any measure taken to minimize confounders thereby improving the validity of the study outcome ? If yes please include in the manuscript to improve the reproducibility of the study .

**Do you want your identity to be public for this peer review?** For information about this choice, including consent withdrawal, please see our Privacy Policy

Reviewer #1: **Yes:** Abimbola Adegoke

Reviewer #2: **Yes:** Taiwo Olufemi Abiona

Reviewer #3: **Yes:** Kenneth Ablordey

Reviewer #4: **Yes:** Sunday Charles Adeyemo

---

## [Editor Report · Decision Letter 1]

25 Jan 2026

PGPH-D-25-01388R1

Application of Machine Learning Approaches to Develop Predictive Models for Diabetes and Hypertension among Bangladesh Adults

Dear Dr. Kibria,

Thank you for submitting your manuscript to PLOS Global Public Health. After careful consideration, we feel that it has merit but does not fully meet PLOS Global Public Health’s publication criteria as it currently stands. Therefore, we invite you to submit a revised version of the manuscript that addresses the points raised during the review process.

Please address the followings in the revision:

Title: Please change Bangladesh adults to Bangladeshi adults.

Use the term “sex” instead of “gender” throughout the manuscript.

Abstract: revise this sentence, particularly the first clause; it is difficult to understand for the lay audience. “Hypertension topped diabetes predictors, while overweight/obesity was the top predictor for hypertension, followed by age and diabetes.”

Section “Potential Factors” in the methods: since the study included both males and females, why is only women’s age? So shouldn’t it be the age of men and women?

We look forward to receiving your revised manuscript.

Kind regards,

Saifur R. Chowdhury, BScN, MPH, PhD (c)

Academic Editor

Journal Requirements:

1. Please note that PLOS Global Public Health has specific guidelines on code sharing for submissions in which author-generated code underpins the findings in the manuscript. In these cases, all author-generated code must be made available without restrictions upon publication of the work. Please review our guidelines at https://journals.plos.org/globalpublichealth/s/materials-and-software-sharing#loc-sharing-code and ensure that your code is shared in a way that follows best practice and facilitates reproducibility and reuse.
---

## [Editor Report · Decision Letter 2]

3 Feb 2026

Application of Machine Learning Approaches to Develop Predictive Models for Diabetes and Hypertension among Bangladeshi Adults

PGPH-D-25-01388R2

Dear Dr. Kibria,

We are pleased to inform you that your manuscript 'Application of Machine Learning Approaches to Develop Predictive Models for Diabetes and Hypertension among Bangladeshi Adults' has been provisionally accepted for publication in PLOS Global Public Health.

Best regards,

Saifur R. Chowdhury, BScN, MPH, PhD (c)

Academic Editor